# Statins in High Cardiovascular Risk Patients: Do Comorbidities and Characteristics Matter?

**DOI:** 10.3390/ijms23169326

**Published:** 2022-08-18

**Authors:** Enrica Rossini, Federico Biscetti, Maria Margherita Rando, Elisabetta Nardella, Andrea Leonardo Cecchini, Maria Anna Nicolazzi, Marcello Covino, Antonio Gasbarrini, Massimo Massetti, Andrea Flex

**Affiliations:** 1Internal Medicine, Università Cattolica del Sacro Cuore, Largo Francesco Vito 1, 00168 Roma, Italy; 2Cardiovascular Internal Medicine Unit, Fondazione Policlinico Universitario A. Gemelli IRCCS, Largo Agostino Gemelli 8, 00168 Roma, Italy; 3Emergency Department, Fondazione Policlinico Universitario A. Gemelli IRCCS, 00168 Roma, Italy; 4Department of Medical and Surgical Sciences, Fondazione Policlinico Universitario A. Gemelli IRCCS, Largo Agostino Gemelli 8, 00168 Roma, Italy; 5Department of Cardiovascular Sciences, Fondazione Policlinico Universitario A. Gemelli IRCCS, Largo Agostino Gemelli 8, 00168 Roma, Italy

**Keywords:** statin, cardiovascular disease, atherosclerosis, LDL-cholesterol

## Abstract

Atherosclerotic cardiovascular disease (ASCVD) morbidity and mortality are decreasing in high-income countries, but ASCVD remains the leading cause of morbidity and mortality in high-income countries. Over the past few decades, major risk factors for ASCVD, including LDL cholesterol (LDL-C), have been identified. Statins are the drug of choice for patients at increased risk of ASCVD and remain one of the most commonly used and effective drugs for reducing LDL cholesterol and the risk of mortality and coronary artery disease in high-risk groups. Unfortunately, doctors tend to under-prescribe or under-dose these drugs, mostly out of fear of side effects. The latest guidelines emphasize that treatment intensity should increase with increasing cardiovascular risk and that the decision to initiate intervention remains a matter of individual consideration and shared decision-making. The purpose of this review was to analyze the indications for initiation or continuation of statin therapy in different categories of patient with high cardiovascular risk, considering their complexity and comorbidities in order to personalize treatment.

## 1. Introduction

Statins are commonly used drugs in patients at high cardiovascular risk. These drugs reduce serum low-density lipoprotein cholesterol (LDL-C), which is involved in the pathogenesis of cardiovascular disease [1]. Proper treatment of patients with hypercholesterolemia begins with the notion that not all patients are the same and that treatment must be individualized. The first step is to determine the patient’s overall cardiovascular risk [2] (Figure 1). Depending on each patient’s specific risk category, specific therapeutic targets for LDL cholesterol need to be achieved [2]. In addition to cardiovascular risk, individual patient characteristics and possible adverse effects of drugs in specific patient categories must also be considered (Figure 1). In addition, statins vary in their chemical composition (Figure 2), pharmacokinetics, and potency of lowering LDL cholesterol [3].

### 1.1. Are Statins All the Same?

Statins are the drug of choice for the treatment of hypercholesterolemia to lower LDL cholesterol. They act principally in the liver by competitively inhibiting 3-hydroxy-3-methylglutaryl-CoA reductase activity. The pharmacological activity determines the decrease in intracellular cholesterol concentration, which leads to an increase in the expression of LDL receptors on the surfaces of hepatocytes. Increased LDL receptor expression leads to increased uptake of LDL-C in the blood, resulting in lower plasma concentrations of LDL-C and other apolipoprotein B-containing lipoproteins, including triglyceride-rich particles [1]. Although all statins act by the same mechanism of action, they differ in chemical composition and pharmacokinetics, affecting treatment and adverse effects. Lovastatin, pravastatin, and simvastatin are fungal-derived 3-hydroxy-3-methylglutaryl coenzyme A (HMG-CoA) reductase inhibitors, while atorvastatin, cerivastatin, fluvastatin, pravastatin, pitavastatin, and rosuvastatin are fully synthetic compounds [4]. Atorvastatin, fluvastatin, lovastatin, and simvastatin are relatively lipophilic compounds, while pravastatin and rosuvastatin are more hydrophilic (Figure 3) [5]. Lipophilic drugs are more susceptible to oxidative metabolism by the CYP450 system (cytochrome P450) [6]. Due to the risk of drug interactions that inhibit CYP450, statins metabolized by the CYP450 system are more likely to cause muscle damage; this leads to increased blood statin levels, which lead to an increased risk of toxic effects [7]. Several studies have compared lipophilic and hydrophilic statins in the clinical setting [8,9]. Lipophilic statins reduce adenosine triphosphate (ATP) production, which theoretically increases myocardial stunning after ischemia and worsens myocardial function after reperfusion [10]. This was demonstrated in an animal study in which lipophilic statins exacerbated myocardial shock and decreased tissue ATP following coronary reperfusion, whereas hydrophilic statins did not show these effects. A meta-analysis concluded that lipophilic statins have risks of major adverse cardiovascular events, myocardial infarction, and all-cause mortality comparable with hydrophilic statins [11].

The largest study (involving the most patients, drugs, and doses) was the CURVES study [13], which compared doses of atorvastatin (10, 20, 40, and 80 mg), simvastatin (10, 20, and 40 mg), pravastatin (10, 20 and 40 mg), fluvastatin (20 and 40 mg) and lovastatin (20, 40 and 80 mg). This was a multicenter, randomized, open-label, 8-week, parallel-group study of 534 hypercholesterolemic patients with LDL values above 160 mg/dL and triglycerides above 400 mg/dL. Atorvastatin 10, 20, and 40 mg resulted in greater LDL-C lowering than equivalent doses of simvastatin, pravastatin, lovastatin, and fluvastatin (238%, 246%, and 251%). Atorvastatin 10 mg produced LDL-C lowering comparable to simvastatin 10, 20 and 40 mg, pravastatin 10, 20 and 40 mg, lovastatin 20 and 40 mg, and fluvastatin 20 and 40 mg. Atorvastatin 10, 20, and 40 mg resulted in greater reductions in total cholesterol compared to mg-equivalent doses of simvastatin, pravastatin, lovastatin, and fluvastatin. All reductase inhibitors tested were similarly tolerated.

### 1.2. Cardiovascular Risk Category and LDL-C Target

Treatment goals have been defined according to different risk categories (they apply to primary and secondary prevention; treatment must always be combined with lifestyle changes) [2] (Table 1).

## 2. Statins and Cardiovascular Diseases

### 2.1. Acute Coronary Syndrome (ACS)

Randomized clinical trials have demonstrated the efficacy and safety of statins for the primary and secondary prevention of cardiovascular disease [14]. In the context of acute coronary syndrome, other studies have shown that loading doses of statins can attenuate the inflammatory cascade and promote coronary artery vulnerability [15,16] by reducing macrophage and cholesteryl ester levels and increasing collagen and smooth muscle cells [17,18]. Plaque rupture activates the thrombotic cascade: statins mitigate this event by inhibiting platelet aggregation and maintaining a balance between prothrombotic and fibrinolytic mechanisms [17,19,20]. These non-lipid properties of statins may explain the early and significant reduction in cardiovascular events reported in several clinical trials. Several studies and systematic reviews have examined the effect of loading doses of statins before and after percutaneous coronary intervention (PCI) [21,22,23,24,25]. These studies suggest that perioperative myocardial infarction (MI) may be reduced [26]. In addition, statin pretreatment has also been shown to reduce the risk of contrast-induced acute kidney injury after coronary angiography or interventional therapy [27]. Therefore, early initiation or continuation of high-dose statin therapy is recommended for all ACS patients without contraindications or a clear history of intolerance, regardless of initial LDL-C levels [2].

Statin therapy is underused in chronic kidney disease (CKD) patients with ACS [28], although statins have been shown to be safe even in advanced CKD. CKD is associated with short- and long-term adverse events in patients with ACS [29]. In the setting of ACS with rupture, plaque instability, increased inflammatory status, and a prothrombotic environment, CKD patients should also be prescribed statin therapy regardless of their estimated glomerular filtration rate (eGFR) levels [30].

Another risk group for cardiovascular events is HIV-infected individuals, who are less likely to have lower LDL-C after ACS than non-HIV-infected individuals [31]. This may be due to chronic HIV-related infection and inflammation leading to persistent immune activation and use of antiretroviral therapy, which may lead to disturbances in lipid and glucose metabolism [32]. After the initial ACS, HIV-infected individuals are more likely to experience recurrent acute coronary events than non-HIV-infected individuals, but HIV-positive patients are often prescribed less effective or lower doses of statins. Appropriate statin strengths should be prescribed for HIV-infected individuals with attention to potential drug–drug interactions [33].

### 2.2. Peripheral Arterial Disease (PAD)

Patients with peripheral arterial disease, including asymptomatic patients, are at increased risk of death, myocardial infarction, and stroke [34]. In a large cohort study of Danish patients, the authors found that PAD could be considered a risk equivalent for coronary artery disease (CAD) even in the absence of diabetes. This study showed that the combination of PAD and MI was associated with the highest cardiovascular risk [35]. Therefore, clinicians should actively assess and manage cardiovascular risk factors in patients diagnosed with PAD [36], as they do in patients with established CAD. A study by Foley and colleagues compared high-intensity (HI) and low- or moderate-intensity (LMI) statin therapy in patients with PAD undergoing peripheral angiography and/or surgery [37]. Results showed that HI statin treatment provided a mortality benefit over statin LMI treatment in patients with PAD, despite similar baseline LDL levels between groups, and a reduction in major adverse cardiovascular events (MACE) [38]. HI statin therapy has potent LDL-lowering effects and may confer additional benefits through pleiotropic mechanisms associated with regression and plaque stabilization [39,40]. In addition, high-intensity statin use at PAD diagnosis was associated with significantly lower limb loss and mortality compared with low-intensity statin users and patients receiving antiplatelet therapy alone [41]. Therefore, patients with PAD are at very high risk and should be treated according to the recommendations of the European Society of Cardiology (ESC) guideline [2] to ensure aggressive secondary prevention.

### 2.3. Heart Failure

Several observations suggest that statins may be an effective treatment for heart failure (HF) [42]. Small studies have shown that statins improve endothelial function [27] and reduce plasma proinflammatory cytokine levels [28] in patients with CAD and hyperlipidemia. Statins can directly exert antioxidant [43], anti-hypertrophic [44], and anti-fibrotic effects on the myocardium and alter immune function [45], macrophage metabolism, and cell proliferation, in contrast to changes in low-density lipoprotein cholesterol concentrations [46]. Experimental evidence also suggests that statins counteract sympathetic upregulation in acute and chronic heart failure by reducing plasma norepinephrine levels and reducing renal sympathetic nervous system activity [47].

A study by Khush and colleagues showed that high-dose statins not only reduced the rate of major cardiovascular events in high-risk patients, but also hospitalizations in patients with stable coronary artery disease [48]. Statins appear to reduce the development of heart failure, at least in part, by lowering blood lipids and other anti-atherothrombotic mechanisms; in fact, heart failure progression is strongly associated with recurrent ischemic events, and statins can promote stabilization of sclerotic plaques, reducing myocardial necrosis, maintaining myocardial viability and improving ventricular function [49].

A meta-analysis of unpublished data from major randomized trials showed that statins modestly reduced the risks of non-fatal HF hospitalization and a composite of non-fatal HF hospitalization and HF death with no demonstrable difference in risk reduction between those who suffered an MI or not [50]. However, according to the 2019 ESC/European Atherosclerosis Society (EAS) Dyslipidemia Guidelines [2], statins are not recommended for cholesterol-lowering therapy in patients with moderate to severe symptomatic heart failure (New York Heart Association (NYHA) Class III-IV). However, for those already taking statins to prevent CAD [35], continued use may be considered. These recommendations are based primarily on results from two large outcome studies, Controlled Rosuvastatin Multinational Trial in Heart Failure CORONA [51] and the GISSI-HF trial [52]: in both studies, no significant reduction in the primary combined mortality/morbidity endpoint was observed in actively treated patients.

In conclusion, statins reduce the risk of new-onset heart failure in the medium to long term, albeit modestly, well below the comparable benefit on CAD outcomes [53]. Therefore, statins should be continued in heart failure with reduced ejection fraction (HFrEF) patients already receiving statins for coronary artery disease or hyperlipidemia, whereas initiation of statins is not recommended for most patients with chronic heart failure.

### 2.4. Cardiac Valvulopathies

Calcified aortic stenosis (AS) is mediated by a chronic inflammatory process that shares many similarities with atherosclerosis [54,55], given its clinical association with hypercholesterolemia and coronary artery disease and its association with atherosclerosis. Histological similarities have suggested that cessation of statin therapy or even progression induces regression of calcified aortic stenosis [56]. However, randomized clinical trials have not demonstrated that lipid lowering prevents AS progression [57]. In fact, the Simvastatin and Ezetimibe in Aortic Stenosis (SEAS) [58], Scottish Aortic Stenosis and Lipid Lowering (SALTIRE) [59] and the Aortic Stenosis Progression Observation: Measuring Effects of Rosuvastatin (ASTRONOMER) [60] trial studies concluded that lipid lowering may be a treatment to slow AS progression. A review by Thiago and colleagues suggests that the effect of statins on aortic stenosis is uncertain [61]. Therefore, lipid lowering can only have a significant effect in patients with primary or secondary hyperlipidemia; in this case, LDL would be the main driver of disease progression. Valve lesions in the early stages of AS have been shown to resemble lipid-laden atheromatous plaques, leading to irreversible mineralized osteogenic bone formation in later stages, followed by the differentiation of interstitial valve cells into osteoblasts [62]. Experimental models suggest that statin treatment slows the differentiation of interstitial valve cells into bone [63]. A study by Greve and colleagues [55] suggests that lowering blood lipids before osteoblast formation may be the best treatment for altering the natural history of AS. They showed that targeting hyperlipidemia early in the disease prevented AS progression [55]. Therefore, statins may be more useful in the early stages of aortic stenosis than during moderate or severe stenosis. Under no circumstances is it recommended to initiate lipid-lowering therapy to slow progression of aortic stenosis in patients with non-CAD aortic stenosis without other indications for use [2].

### 2.5. Stroke

The use of statins in patients with ischemic stroke improves post-stroke outcomes [64,65], reduces the risk of recurrent ischemic stroke [66] and is, therefore, recommended for patients with a history of ischemic stroke [67] or transient ischemic attack (TIA) [2]. There are conflicting data on the effect of statins on the risk of intracerebral hemorrhage (ICH). A systematic review and meta-analysis of data from 23 randomized trials and 19 observational studies [68] and a further meta-analysis of 31 randomized controlled trials [69] suggest that outpatient statin use does not increase the risk of bleeding. On the other hand, in the SPARCL study (a prospective, randomized, double-blind clinical trial), treatment with atorvastatin (80 mg/day) reduced bleeding in patients with a history of transient ischemic attack (TIA) or recent stroke [70], but a post hoc analysis showed an increase in the number of stroke-bleeding patients receiving treatment (55 in the atorvastatin group and 33 in the placebo group) [70]. A secondary analysis found that statin therapy, increasing age, and ICH as eligible strokes for included studies were factors associated with late-onset ICH. This has led many clinicians to be cautious about prescribing statins to patients with ICH following these findings. Markov analysis concluded that statin avoidance should be considered in patients with a history of ICH, especially in the setting of lobar localization [71]. The association between statin therapy and increased presence and number of microbleeds (MB), which are frequently observed in cerebral amyloid angiopathy (particularly cortico-subcortical microbleeds), was investigated, with the conclusion that statin use in patients with ICH is independently associated with MB, especially csMB [72]. It is unclear whether statin use should be continued in patients with ICH, but a retrospective study suggests that continued statin use after ICH may be associated with early neurological improvement and may reduce mortality within 6 months, which may be due to immunomodulation as a pleiotropic effect of statins [72]. There are no data to suggest that susceptibility to ICH is dose-dependent.

## 3. Statins in Special Populations

### 3.1. Statins and Elderly People

Biologically, people over the age of 75 are a very heterogeneous group, often with disabilities, comorbidities, and multiple concomitant medications. In primary prevention, a person’s life expectancy must be considered. Prevention may be useless if started too late, as in patients with dementia [73] or advanced heart [52] or renal failure [74]. Results of randomized controlled trials and observational studies in younger patients and more than 75 year-aged subgroups support the treatment of secondary prevention of ASCVD. According to a meta-analysis of randomized statin trials involving 14,483 participants over the age of 75, statin therapy was associated with a 15% reduction in the rate of major vascular events [75]. Conversely, the evidence from primary prevention studies is less clear [76,77]. A systematic review by Ravnskov and colleagues showed that low cholesterol is associated with poorer outcomes in older age [78], raising concerns and criticism about the importance of statins in older adults. In the frail elderly, low serum cholesterol may be an indicator of changes in cholesterol metabolism [75], a marker of terminal decline [75], or a marker of subclinical diseases such as cancer [75]. Frail people with comorbidities and polytherapy may also be more prone to adverse effects and drug interactions: adverse muscle effects could promote sarcopenia and predispose to frailty, falls and morbidity, but there is no hard evidence of this in the available studies [79], and there is no evidence that very low cholesterol could be associated with cognitive impairment [80]. In conclusion, statin therapy is generally safe and well-tolerated in these patients, and therapy should be continued for those aged more than 75 years. If a patient is considered to be at increased risk of developing ASCVD, then primary prevention therapy needs to be initiated in older age [76]. Finally, the recent ESC 2021 cardiovascular disease prevention guidelines provide a single cut-off point for identifying “elderly” as 70+ rather than 75, but also emphasize that all of these age intervals are relatively arbitrary, and biological age affects this threshold in clinical practice. In addition, according to the latest ESC guidelines, starting statin therapy for primary prevention at age 70 in patients at very high cardiovascular risk may be considered, subject to other factors such as risk modifiers, frailty, estimated benefit over the life course, comorbidities and patient preferences. Regarding LDL-C targets, there is insufficient evidence to support primary prevention targets in elderly patients, but if there is significant renal impairment and/or the possibility of drug–drug interactions, it is recommended to start with low-dose statins [81].

### 3.2. Statins and Young People

The 2013 American College of Cardiology (ACC)/American Heart Association (AHA) blood cholesterol management guidelines recommend statin therapy for primary prevention of ASCVD in individuals ≥ 21 years of age with LDL-C ≥ 190 mg/dL. Statins may be considered in patients up to 75 years of age with diabetes or an estimated 10-year ASCVD risk of ≥7.5%, and in those with an estimated 10-year risk of 5% to 7.5% [81]. Due to the lack of evidence from RCTs, this guideline does not recommend statin therapy for adults younger than 40 years with LDL-C < 190 mg/dL. Recently, the AHA, ACC, and several other health organizations collaborated on the 2018 cholesterol guidelines [82]. These guidelines are consistent with the previous 2013 ACC/AHA guideline; in addition, for individuals 20 to 39 years of age with LDL-C < 190 mg/dL, a lifetime risk assessment is recommended, with particular attention to lifestyle modifications to consider the risk of premature ASCVD. Individuals with a family history and LDL-C > 160 mg/dL receive statin therapy to reduce ASCVD risk. A novelty of the 2018 guidelines is the recommendation to test the coronary artery calcium score (CAC) in persons at intermediate risk (10-year risk ≥ 7.5% ≤20%) when uncertain about the decision to initiate statin therapy [12]. The development of coronary atherosclerosis is a lifelong process with atherosclerotic changes evident in early adulthood [83], and statin therapy has been shown to reverse coronary atherosclerosis [84]. There is concern about starting statin use because of the side effects associated with myalgia and arthralgia; in addition, there has been a reported risk of developing diabetes with statin use, although the risk is small and the potential benefit appears to be significantly greater [85]. Additionally, it is unclear whether long-term statin use leads to an increased risk of diabetes. Randomized trials have also failed to demonstrate cognitive impairment problems associated with statin use [86]. On the other hand, women wishing to have children should avoid statins due to their possible teratogenic effects. To improve risk stratification to help decide when and whether to start statin use in young adults, we are trying to identify risk markers such as CAC [87]. A recent analysis of the Coronary Artery Risk Development in Young Adults (CARDIA) study showed that CAC is a strong predictor of future CAD in young adults. Among 3043 CARDIA participants aged 32 to 46 years (mean age 40.3 years), 10.2% had CAC at baseline, and those with any CAC had a 5-fold and 3-fold increase in coronary heart disease events and cardiovascular deaths, respectively, after adjustment for demographics, risk factors, and cardiovascular medications [88]. Although not yet clinically available, polygenic risk assessments for ASCVD outcomes have been developed and show promise for improved risk stratification [89]. Further studies are needed to assess the effectiveness of the use of polygenic risk scores and CACs in the decision to introduce statins. In conclusion, for patients in their 20s with increased lifetime risk, the focus should be on a healthy lifestyle, and statin use should be limited to those with familial hypercholesterolemia. For individuals in their 30s at increased lifetime risk, low- to moderate-intensity statin therapy may be offered to patients who place a high priority on reducing future ASCVD risk. In addition, CAC testing can be performed in selected individuals (mainly men in their 30s), although the results should be interpreted with caution given the low prevalence of CAC in this age group.

### 3.3. Statins and Familial Dyslipidemias

Familial hypercholesterolemia (FH), the most common and severe form of congenital hypercholesterolemia, significantly accelerates the onset of ASCVD, primarily coronary artery disease [90]. Statins are first-line therapy for LDL cholesterol in FH, especially in heterozygous patients [91]. FH is undertreated, with more than 80% of FH patients receiving statins failing to achieve LDL-cholesterol goals. This may be related to adherence, genetic differences and tolerance to statins. Current guidelines recommend that children homozygous for FH should be treated as early as possible at the time of diagnosis. Treatment for children who are heterozygous for FH should be initiated at the lowest recommended dose (such as pravastatin 20 mg, atorvastatin 10 mg, and rosuvastatin 5 mg) and up-titrated according to the LDL cholesterol-lowering response and tolerability from 8 to 10 years of age [2,92]. In general, it is recommended to consider FH patients with ASCVD or other major risk factors as very high-risk patients, and patients without previous ASCVD or other risk factors as high-risk patients.

### 3.4. Statins and Cognitive Impairment

Some cases have linked statin use to cognitive impairments, such as short- and long-term memory loss [93], behavioral changes, concentration and attention problems, anxiety, and paranoia [94]. The statins involved are simvastatin, atorvastatin, and rosuvastatin. The underlying mechanism of cognitive impairment is based on the relationship between cholesterol and myelin; myelin is composed of cholesterol, and statins reduce the de novo synthesis of cholesterol [95]. Changes in myelination can also lead to changes in nerve signaling pathways that can lead to cognitive decline. Another mechanism may involve oxidative stress and mitochondrial function [96]. Statins lower coenzyme Q10 levels by inhibiting mevalonate synthesis. In turn, coenzyme Q10 is responsible for the correct function of mitochondria, is involved in the production of adenosine triphosphate and has antioxidant functions. Therefore, statins will determine changes in mitochondrial function, increased oxidative stress, and thus cognitive deterioration. On the other hand, high cholesterol levels are associated with an increased risk of Alzheimer’s disease [97,98]. In this case, the protective effect of statins would be attributed to the pleiotropic effects of these drugs (reduced endothelial dysfunction, increased endothelial nitric oxide production, anti-inflammatory, antioxidant, and antithrombotic properties, vascular generation and other angioprotective properties) [99,100].

If patients at high cardiovascular risk develop cognitive impairment due to statin use and require continued lipid-lowering therapy, the use of less lipophilic statins that cannot cross the blood–brain barrier, such as pravastatin and rosuvastatin, should be considered [101]. Despite a 2012 U.S. Food and Drug Administration (FDA) warning about the potential adverse effects of statins on cognition, a systematic review and meta-analysis of randomized controlled trials found that statin therapy had no significant effect on cognitive function [86]. Therefore, these results call into question the aforementioned FDA warning that statins may have negative effects on cognition.

### 3.5. Statins and Gender

Statins are recommended for reducing primary and secondary cardiovascular disease (CVD) risk and all-cause mortality in women and men, and are based on LDL cholesterol levels, presence of atherosclerosis and/or diabetes, and cardiac vascular risk classes. However, these factors do not take into account whether atherosclerotic plaque development is the same in both sexes [102]. Atherosclerosis is defined as an inflammatory disease, and the predisposing factors for the development of the disease are due to immune activation regulated by genes on Y chromosome [4]. Genetic variations in statin transporters and cytochromes lead to differential responses to statins and different side effects associated with myotoxicity [103,104]. Because these studies were not sex-disaggregated, it is unclear whether the interaction between genetic variation and sex affects the efficacy and side effects of statins. In addition, statins and estrogens share some metabolic pathway enzymes and transporters, which means that they can compete with each other, which may lead to drug–hormone interactions [105]. A sex-specific meta-analysis supported by three primary prevention studies including a large cohort of women was performed: Management of Elevated Cholesterol in the Primary Prevention Group of Adult Japanese (MEGA) study [106], Justification for the Use of Statins in Prevention: An Intervention Trial Evaluating Rosuvastatin (JUPITER) study [107], and Heart Outcomes Prevention Evaluation-3 (HOPE-3) study [108]. Despite the reduction in secondary prevention, statins for primary prevention did not significantly reduce cardiovascular disease [109]. Furthermore, it can be demonstrated that all-cause mortality in women is not affected by statin treatment in primary or secondary prevention [110]. However, a more recent meta-analysis of data from 22 trials of statin therapy versus control studies and five trials of more intensive versus less intensive statin therapy showed that, in men and women with an equivalent risk of cardiovascular disease, statin therapy was of similar efficacy for the prevention of major vascular events [111]. Currently, oral contraceptives do not appear to increase adverse coronary events and can be used in women with acceptable total cholesterol levels after baseline lipid analysis [112]. However, other contraceptives should be recommended for women with hypercholesterolemia (LDL-C > 160 mg/dL (>4 mmol/L)) or multiple risk factors and women at high risk for thrombotic events [113]. In addition, guidelines recommend statins for primary and secondary prevention in women at high cardiovascular risk, with the same indications and goals as those for men [2].

### 3.6. Statins and Pregnancy

Cardiovascular disease complicates pregnancy due to the larger number of women with cardiovascular risk factors and the older maternal age [114]. Women should use birth control while receiving statins, and women who are breastfeeding should not take lipid-lowering drugs until they stop breastfeeding. In the event of an unintended pregnancy, women should discontinue statins despite the cardiovascular risk [115]. However, conflicting data exist: case reports and retrospective reviews [116] suggesting an association between statins and teratogenic effects, and a multicenter, prospective, observational study of pregnant women exposed to statins in early pregnancy. Research has shown that despite an increase in preterm birth, a clear association between statin use during pregnancy and congenital abnormalities has not been found [117]. However, statins should be avoided during pregnancy and discontinued prior to conception due to limited data and information quality [118].

### 3.7. Statins and Perioperative Period

Cardiovascular complications are the most common cause of perioperative morbidity and mortality after major noncardiac surgery. It is currently recommended that patients on statins as maintenance therapy continue to use statins in the perioperative period [119]. Perioperative patients may be at increased risk of drug interactions, and analgesic use may mask symptoms associated with adverse events, such as myalgia. On the other hand, clinical trials in the perioperative period have reported neither major adverse effects of statins nor cases of rhabdomyolysis and liver dysfunction [120]. Evidence suggests that patients who have been taking statins long-term before surgery may have worse outcomes if they stop taking them [121]. In particular, patients with acute coronary syndrome who discontinue statins are at higher risk for postoperative cardiovascular complications [122,123], as are patients undergoing vascular surgery [121]. Possible mechanisms of increased cardiovascular risk following statin withdrawal have been attributed to eNOS-mediated reductions in vasodilation [124], increases in markers of aWBC adhesion, and reductions in tissue plasminogen activator [125]. On the other hand, statins can be used to reduce perioperative risk: three high-quality prospective randomized controlled trials examined cardiovascular outcomes and MI and showed how statin therapy is associated with improved postoperative cardiovascular outcomes [126,127,128]. Most studies involved vascular surgery patients who were at high risk for postoperative cardiac complications due to the high incidence of clinically occult cardiac ischemia. Observational studies have shown that statin therapy reduces perioperative myocardial ischemia, 30-day and late cardiovascular events, and improves long-term survival [129,130]. Both the American College of Cardiology/American Heart Association (ACC/AHA) [131] and the European Society of Cardiology [132] recommend that patients on long-term statins be reinstated as soon as possible after surgery. Patients with non-coronary atherosclerosis should receive statin therapy for secondary prevention independent of non-cardiac surgery. In addition, the European Society of Cardiology recommends starting statins in patients undergoing high-risk surgery, preferably between 30 days and at least 1 week before surgery [2]. The ACC/AHA guidelines state that statins are indicated for patients with vascular surgery with or without clinical risk factors and should be considered for moderate-risk surgery in patients with at least one clinical risk factor. Therefore, perioperative statin therapy is safe and beneficial in reducing perioperative morbidity and mortality, with the greatest benefit in patients at higher risk for cardiovascular complications.

### 3.8. Statins and Chronic Renal Failure

Patients with chronic kidney disease have a higher risk of dying from cardiovascular disease than the general population, and cardiovascular disease is the leading cause of death in patients with kidney disease not receiving dialysis [133]. Several clinical studies have demonstrated the safety and efficacy of statins in patients with CKD, and a previous systematic review found a reduced risk of cardiovascular death in patients with chronic kidney disease not yet deserving of dialysis [134]. A meta-analysis of 28 randomized trials showed that the use of statins reduced the risk of a first major vascular event by 21% [135]. The 2013 Kidney Disease: Improving Global Outcomes (KDIGO) clinical practice guidelines recommend statins with or without ezetimibe for primary prevention of cardiovascular disease in patients over 50 years of age with eGFR < 60 mL/min/1.73 m^2^. In adults < 50 years of age with chronic kidney disease, known coronary artery disease, diabetes, previous ischemic stroke, or estimated cardiovascular risk > 10% at 10 years of age, statin therapy should be initiated. These recommendations do not apply to patients with end-stage renal disease (dialysis or undergoing kidney transplantation) [136]. According to the 2014 Kidney Disease Improvement Global Outcomes Lipid Task Force, initiation of statin therapy in chronic dialysis patients is not recommended [137] because statins provide little benefit in preventing cardiovascular disease and death from any cause in patients starting dialysis, based on several large randomized controlled trials [138,139,140] and systematic reviews [74]. Evidence suggests that patients on dialysis may have different pathogenesis of arterial lesions, and the lesions are so severe that statins are unlikely to significantly reduce them [141,142]. However, one study suggests that statin use may be beneficial in hemodialysis patients with elevated LDL [143] or atherosclerotic cardiac events [144]. Another study assessed the association between initiation of statin use and the development of cardiovascular disease in maintenance dialysis patients [145]. Initiation of statin therapy is associated with an overall increased risk of cardiovascular disease, and the risk varies with the nature of the statin and treatment adherence. For example, hydrophilic statins were associated with a lower risk of cardiovascular disease compared with the use of lipophilic statins. Reduced all-cause mortality was associated with initiation of statin use, and there was no heterogeneity between types of statin use. Initiation of statin use in maintenance dialysis patients was associated with a higher subsequent risk of CVD, but the additional risk associated with hydrophilic statins was lower than that with lipophilic statin use. Lipophilic statins reduce ATP production and theoretically increase myocardial stunning after ischemia and worsen shortening of reperfusion. Therefore, current guidelines recommend the use of statins in patients with KDIGO stages 3–5 renal disease who are not receiving dialysis, and consideration of continued therapy in patients starting dialysis, especially those with ASCVD. Initiation of statin therapy in CKD patients without ASCVD requiring dialysis is not recommended. Finally, considering that many statins are metabolized by the liver and some are metabolized in patients with advanced renal disease (GFR < 30 mL/min/1.73 m^2^), no dose adjustment of statins is required in patients with mild to moderate chronic kidney disease. However, this does not apply to atorvastatin, which is administered in doses of up to 80 mg regardless of CKD stage [146] because it is not excreted through the kidneys [147]. In contrast, other statins require dose reductions in patients with advanced CKD.

### 3.9. Statins and Muscle Disease

People with muscle disease often avoid statins because of known side effects. The cause of statin-induced myopathy is unclear, although several mechanisms have been proposed, including increased oxidative stress [148], activation of the atrogynous-1 muscle wasting pathway [149], and increased susceptibility to RyR1-induced Ca2+ leak in a malignant hyperthermia mouse model [150]. Although statins may be myotoxic, one study suggests that statins are highly beneficial in skeletal muscle affected by underlying conditions such as ischemia, oxidative stress, and inflammation. Statins reduce oxidative stress, inflammation and fibrosis, all processes associated with functional muscle decline in muscular dystrophy, especially Duchenne muscular dystrophy (DMD). Simvastatin has been shown to improve muscle strength and fatigue resistance in DMD mice [151]. In DMD, the leading cause of death is heart failure. In another study, treatment of mdx mice with low to moderate human-equivalent doses of simvastatin significantly improved total left ventricular function in vivo in the short and long term. The main benefits of simvastatin include sustained improvement in diastolic function, increased heart rate variability due to increased parasympathetic activity, and prevention of cardiac fibrosis [152]. However, the 2019 ESC/EAS guidelines recommend against starting statin therapy if baseline CK > 4 ULN [153].

### 3.10. Statins and HIV-Infected Population

HIV-infected patients have a 1.5- to 2-fold increased risk of cardiovascular disease, and the drugs most commonly used to reduce this risk are statins [154]. There are no guidelines for statin therapy specifically for the primary prevention of cardiovascular disease in HIV-infected patients. Several studies have examined the effect of statins on mortality and have shown a reduction in statin-related mortality in the HIV population and a small and non-statistically significant reduction in all-cause mortality [2,155]. The risk of toxicity is increased when statins are co-administered with ritonavir-enhanced protease inhibitors and less frequently with other anti-retroviral therapies [156,157]. In contrast, there were no clinically meaningful interactions between statin therapy and an integrase inhibitor with or without cobicistat [158,159]. Studies on the association between statins and the onset of diabetes in HIV-infected patients are conflicting, with some studies showing an increased risk [160] and others not [161,162]. However, with regard to muscle toxicity, some studies have reported increased muscle toxicity in HIV-infected adults compared with HIV-uninfected adults [163], although other studies have failed to demonstrate this association [164]. In a large aging cohort study of veterans, HIV-infected statin users had a lower risk of acute liver injury and death compared with non-statin users [165]. Finally, no incidence of dementia associated with statin use has been reported in this population [166]. These drugs usually have few side effects at the appropriate doses of antiretroviral therapy. The trend toward underuse of statin therapy in HIV-infected patients may be related to fear of the aforementioned side effects, but preliminary observational data suggest potential benefits for cardiovascular morbidity and all-cause mortality in routine care.

### 3.11. Statins and Liver Disease

Until recently, statins were considered relatively contraindicated due to concerns about their use in patients with liver disease due to potential hepatotoxicity that could further exacerbate disease progression [167,168]. Despite clear dyslipidemia or cardiovascular disease indications, statins in patients with advanced liver disease have long been underestimated [169]. In fact, later studies have shown that patients with advanced liver failure have no increased risk of severe drug-induced liver disease from taking statins compared with the general population [170], with the exception of atorvastatin and Child–Pugh class C patients, in which statin doses need to be adjusted to avoid high blood levels [171]. Statins have been shown to be beneficial in patients with liver disease: lower portal pressure, improved hepatic sinusoidal endothelial and hepatic microvascular dysfunction, reduced fibrosis, prevention of ischemia/reperfusion injury, safe prolongation of ex vivo liver transplant preservation, reduced sensitivity to endotoxin-mediated liver damage, protection from acute-on-chronic liver failure (ACLF), prevention of liver injury following hypovolemic shock, and prevention/delaying the progression of cirrhosis of any etiology [172,173,174,175,176]. Liver injury from statins is an idiosyncratic reaction that occurs most often in the first few months of use, although a long latency period of about 10 years has been reported. The most common statins are atorvastatin, simvastatin, and fluvastatin [177]. Patients with pre-existing liver disease do not have an increased risk of altered liver enzymes, but decompensated cirrhosis is another condition, as it may increase blood concentrations of drugs, thereby increasing the risk of side effects [178]. Statins should be used in these patients if they have a metabolic or cardiovascular indication, especially if we consider the fact that patients suffering from cirrhosis with dysmetabolic etiology (NAFLD) have strong risk factors for cardiovascular mortality in addition to and sometimes greater than liver-related mortality. In patients with compensated cirrhosis, statins are safe and can be used at regular doses. Atorvastatin was associated with significant hepatotoxicity in 1 in 3000–5000 users, with a 0.3% risk of elevated transaminases, rising to as high as 2.3% in patients receiving high doses (≥80 mg). There are also case reports of atorvastatin-pronounced autoimmune hepatitis [179]. However, there are limited safety data on patients with decompensated cirrhosis. Simvastatin 40 mg once daily was associated with an increased risk of muscle toxicity, while a small study showed that a dose of 20 mg once daily was safe [180]. In a small study of short-term treatment of patients with Child–Pugh B, pravastatin 40 mg once daily was not associated with toxicity [181]. Therefore, these doses of simvastatin and pravastatin may be options for decompensated patients when statins are required. Short-term hepatic mortality is high in patients with Child–Pugh C, and statins are unlikely to alter this [182].

### 3.12. Statins and Diabetes

Statins are the main target of lipid-lowering therapy in patients with DM2. DM itself is an independent risk factor for cardiovascular disease and is associated with a higher risk of cardiovascular disease [183], and every 1 mmol/L LDL-C reduction with statin therapy was associated with a 23% reduction in the 5-year incidence of major cardiovascular events, regardless of initial LDL-C levels or other baseline characteristics, according to a meta-analysis of 14 randomized trials of statins [184]. In recent years, several randomized controlled trials have reported that statin use increases the risk of type 2 diabetes, especially in high-intensity patients and those with multiple risk factors [185]. In the JUPITER study, researchers observed a composite outcome of major cardiovascular events (myocardial infarction, stroke, hospitalization for unstable angina, arterial revascularization, or cardiovascular death) in the rosuvastatin group compared with significantly reduced risk in the placebo group. Diabetes cases were more common in the statin group, especially in patients with multiple risk factors [186]. Subsequent meta-analyses of primary and secondary prevention RCTs for different statins and doses confirmed some excess risk of T2D with statin use [187]. In addition, more intensive statin use has been found to increase the risk of developing diabetes [188,189]. Observational studies have also reported that statin use increases the development of diabetes [190]. Molecular mechanisms that may be involved are discussed, all of which lead to decreased insulin secretion from pancreatic beta cells or increased insulin resistance [191]. Hypothetical mechanisms include: statin-induced inhibition of Ca2+ channels in pancreatic beta cells, leading to a direct reduction in insulin secretion [192], inhibition of HMG-CoA reductase, which also leads to altered insulin sensitivity and loss-of-function genetic polymorphisms [193,194], and expression of glucose protein transporter 4 (GLUT4) and ubiquinone (CoQ10) levels that decrease with changes in adiponectin concentration [195,196]. Furthermore, it was also reported that the effects of statins increase very low-density lipoprotein (VLDL) particle size and reduce LDL and high-density lipoprotein (HDL) particle size, increasing the risk of new-onset diabetes mellitus (NODM) [197,198]. In addition, inhibition of pancreatic β-cell transporter induced altered glucose metabolism [199,200]. A study by Fahim Abbasi et al. found that short-term treatment with high-intensity atorvastatin therapy resulted in increases in insulin resistance accompanied by increases in insulin secretion [201]. The risk of new-onset T2D associated with statin therapy may increase in those individuals who become more insulin-resistant but are unable to maintain compensatory increase in insulin secretion. Fahim Abbasi et al. recruited volunteers without T2D who were eligible for statin therapy for primary prevention of ASCVD. Participants with metabolic syndrome were more insulin-resistant and had higher insulin secretion than those without metabolic syndrome. The mechanism of increase in insulin resistance associated with statin therapy and the cellular mechanisms that could explain the increased insulin secretion are not completely understood. Persons with more severe metabolic syndrome are at higher risk for developing incident T2D due to statin use [202,203]. In that regard, individuals with metabolic syndrome have features such as greater insulin resistance and higher fasting glucose that increase their risk of T2D. Some studies in the literature suggest that long-term statin use could cause weight gain and thereby increase insulin resistance, but in this study, they detected an increase in insulin resistance without an increase in weight gain. Another possible explanation concerns genetics: it could play a potentially negative role leading to increased intracellular cholesterol and risk for T2D and insulin resistance [204]. Individuals with naturally occurring mutations that inhibit HMGCR have low plasma LDL-C levels, but increased intracellular cholesterol levels and a greater risk of T2D [191], while individuals with mutations in the low-density lipoprotein receptor (LDLR) have extreme elevations in plasma LDL-C levels but a decreased prevalence of T2D proportional to the severity of the LDLR mutation [195]. Other proposed mechanisms for how statins may increase insulin resistance include deregulation of intracellular or membrane-bound cholesterol levels, suppression of intracellular levels of isoprenoids, perturbation of insulin signaling pathways, accumulation of free fatty acids and mitochondrial dysfunction [201]. In addition to an increase in insulin resistance, it was noticed that a short-term statin treatment increases insulin secretion, a well-known compensatory response to increases in insulin resistance [205]. The increase in insulin secretion was driven by change in insulin resistance to maintain glucose homeostasis, but in some participants, insulin secretion decreased despite the increase in insulin resistance. This pattern may indicate an inability to compensate for increases in insulin resistance and might be a harbinger of statin-related T2D. Given that statin use is known to reduce cardiovascular risk, experts agree that statin therapy should not be discontinued for fear of an increased risk of diabetes [206,207]. The balance of risks and benefits remains in favor of statin therapy: the risk of diabetes should not be a reason to discontinue treatment, and lifestyle changes should be made to reduce cardiovascular and diabetes risk in statin users [186].

### 3.13. Statins and Cancer

In patients with limited prognosis, the risks of some drugs may outweigh the benefits. Data on the risks and benefits of discontinuing statin therapy in patients with limited life expectancy are lacking [208]. One study showed that in people with a median survival of 7 months and a diagnosis of cancer or no cancer, statin discontinuation is safe, and that discontinuation of statins offers several benefits, such as: statin drugs and drug costs are correspondingly reduced. Survival was not affected when statins for primary or secondary prevention of cardiovascular disease were discontinued in this population. In a systematic review and meta-analysis of observational studies by Zhong et al., positive effects on overall and cancer-specific survival were noted after and before diagnosis [209]. However, statin use may not always be beneficial in individual settings, especially post-diagnostic statin use. Regarding cancer-related mortality, a study observed that statin use in patients with cancer was associated with reduced cancer-related mortality, probably because statins inhibit cholesterol synthesis within cells through the inhibition of HMG-CoA reductase, the rate-limiting enzyme in the mevalonate and cholesterol-synthesis pathway. Many of these downstream products are used for cell proliferation because they are required for key cellular functions such as maintenance of membrane integrity, signaling, protein synthesis, and cell cycle progression [210,211], and disruption of these processes in malignant cells results in inhibition of cancer growth and metastasis [212]. Statins have been implicated in arresting cell cycle progression in cancer cells, resulting in antiproliferative effects, inhibition of important cellular functions in cancer cells, and increased radiosensitization [213,214]. A reduction in plasma cholesterol levels may be a plausible mechanism behind the observed reduction in cancer mortality risk. In fact, fast-growing cancers require a large uptake of extracellular cholesterol, and plasma cholesterol levels are reduced in cancer patients [215,216]. Thus, statin-induced reductions in local synthetic or circulating cholesterol levels can inhibit cancer growth and metastasis and reduce mortality.

### 3.14. Statins and Transplant Patients

Cardiovascular mortality has been consistently high among transplant recipients with functional grafts [217]. Due in part to strong immunosuppression, more than 80% of adult kidney transplant recipients develop dyslipidemia within the first year [218], and patients with pretransplant hyperlipidemia are more likely to have persistent dyslipidemia [219]. There is increasing evidence that the presence of metabolic syndrome (MS) contributes to chronic graft dysfunction, graft loss, new-onset diabetes after transplantation (NODAT), and patient death [220]. The Symphony study examined the course of MS-related laboratory parameters in the first year after transplantation. It is well known that immunosuppressive therapy plays an important role in the development of metabolic complications after transplantation. The prevalence of hypertension, impaired glucose tolerance, and hyperlipidemia is high after transplantation. There were differences in metabolic parameters between the immunosuppressed groups: cyclosporine treatment was associated with the highest levels of uric acid and systolic and diastolic blood pressure, whereas patients treated with sirolimus had the worst lipid control (dose-dependent increases in serum triglycerides and LDL cholesterol). Finally, a possible effect of tacrolimus on the pathogenesis of diabetes cannot be ruled out. Management of dyslipidemia in transplant recipients is comparable to recommendations for patients at high or very high risk for ASCVD, but more attention needs to be paid to the causes of dyslipidemia and possible side effects from drug interactions [2]. Immunosuppressants are metabolized by the cytochrome P450 (CYP450) enzyme system and cleared from cells by the p-gp protein, a multi-transporter for drug resistance. CYP3A4 is especially important because 60% of oxidative drugs, including calcineurin, cyclosporine, tacrolimus, sirolimus and everolimus, are biotransformed by this special enzymatic system [221]. One review evaluated the interaction of statins with tacrolimus and cyclosporine. Studies have shown pharmacokinetic differences between cyclosporine and tacrolimus, particularly in the inhibition of two hepatic transporters: P-glycoprotein and organic anion-transporting polypeptide. Compared with cyclosporine, tacrolimus does not affect these transporters, does not increase statin exposure, and does not increase statin-related adverse events [222]. Clinical practice guidelines suggest that statin dose reduction is required when using tacrolimus. This practice, adopted by some providers, prevents transplant recipients from gaining cardiovascular benefits, especially when escalating or high-intensity doses are required. Clinicians need to be aware that tacrolimus and cyclosporine are not the same in causing interactions with statins. Tacrolimus can be used concomitantly with statins without dose adjustment due to lack of interactions [222]. Therefore, statins should be considered as first-line drugs in transplant patients, starting at low doses and titrated with caution, with attention to potential drug interactions, especially in patients receiving cyclosporine [2].

## 4. Conclusions

Statins remain an effective therapy for the primary and secondary prevention of cardiovascular disease and are pleiotropic. At the same time, there is a tendency to under-prescribe these drugs due to concerns about side effects in different patient populations. This has led to under-prescribing of statins without reaching optimal LDL-C levels in patients’ cardiovascular risk categories. Awareness of the benefits of statin therapy was low, as was knowledge of guideline-recommended eligibility for statin therapy. Physicians should assess the patient for all complexities, determine cardiovascular risk, optimal LDL cholesterol levels, and possible side effects from statin use, and select the most appropriate statin type and dose for the patient. Statin treatment should be personalized rather than under-prescribed, considering the guidelines, but also the complexity and comorbidities of the patient should be considered in order to provided treatment that best balances the risks and benefits (Table 2).

## Figures and Tables

**Figure 1 ijms-23-09326-f001:**
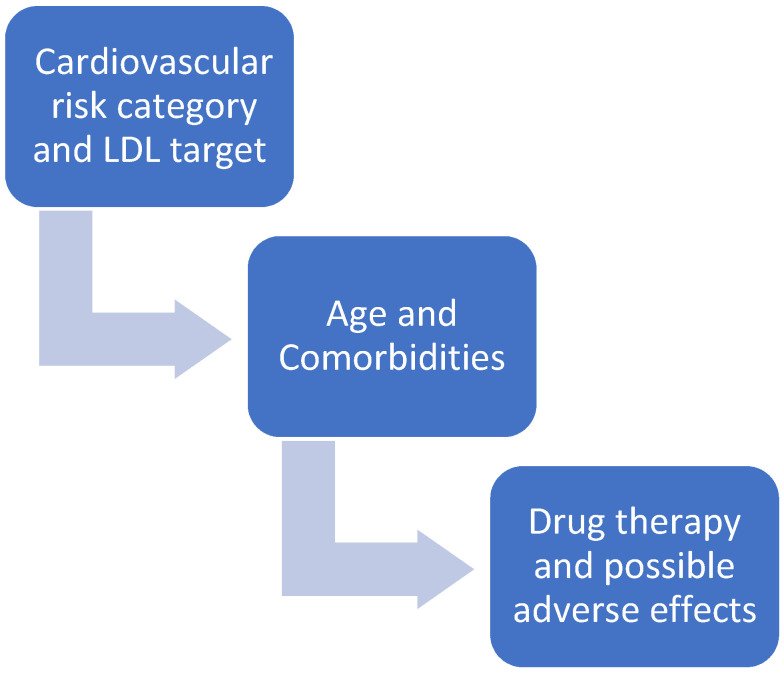
How to decide which statin and what dosage.

**Figure 2 ijms-23-09326-f002:**
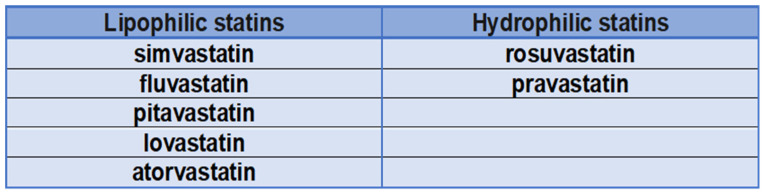
Hydrophilic and lipophilic statins.

**Figure 3 ijms-23-09326-f003:**
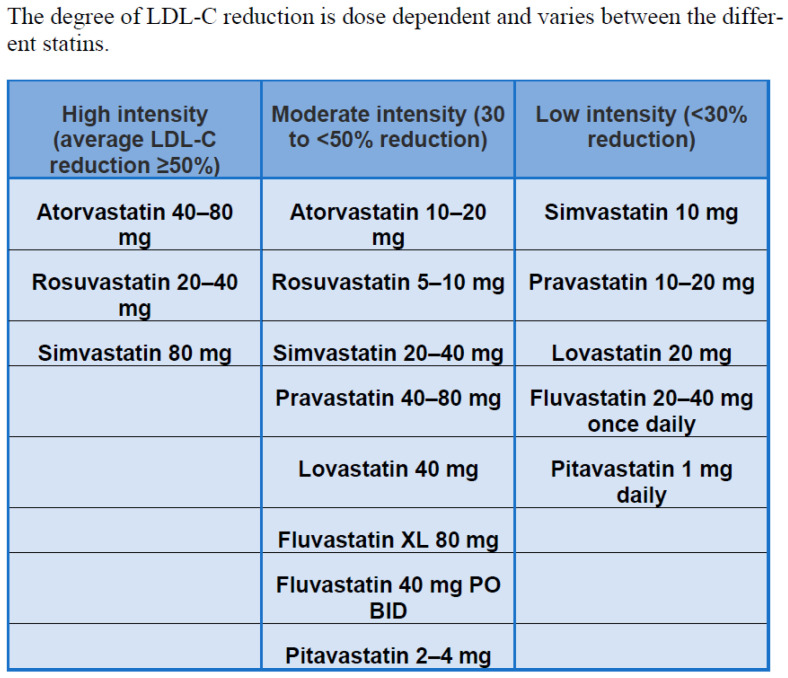
Intensity of statin therapy [12].

**Table 1 ijms-23-09326-t001:** LDL-C target according to cardiovascular risk.

RISK CATEGORY.	WHICH PATIENT?	LDL-C TARGET
Very high-risk patients (10-year risk of cardiovascular mortality > 10%)	Atherosclerotic cardiovascular disease (ASCVD) documented clinically or by imaging (acute coronary syndrome, stable angina, coronary revascularization, stroke or transient ischemic attack, peripheral arterial disease). Imaging documented ASCVD, including findings known to be relevant to the development of future clinical events, such as Diabetes mellitus (DM) with end-organ damage (microalbuminuria, retinopathy, and neuropathy) or at least 3 CV (cardiovascular) risk factors or early-onset type 1 diabetes that has been present for more than 20 years. Severe chronic kidney disease (eGFR < 30 mL/min/1.73 m^2^).	LDL < 55 mg/dL or reduce LDL by at least 50% compared to baseline levels
Very very high-risk patients	Very high-risk patients who experience a second vascular event within 2 years of the first during therapy with statins at the highest tolerable dosage.	LDL < 40 mg/dL
High-risk patients (10-year risk of cardiovascular mortality 5–10%)	Particularly high individual risk factors, such as total cholesterol > 310 mg/dL (>8 mmol/L), LDL-C > 190 mg/dL (>4.9 mmol/L) or blood pressure ≥ 180/110 mmHg. Familial hypercholesterolemia without other CV risk factors. Diabetes mellitus without end organ damage, but present for at least 10 years or in conjunction with another CV risk factor. Chronic moderate kidney disease (eGFR 30–59 mL/min/1.73 m^2^).	LDL < 70 mg/dL or reduce LDL values by at least 50% compared to the initial ones
Moderate risk patients (10-year risk of cardiovascular mortality > 1% <5%)	Diabetes in young subjects (T1DM < 35 years, T2DM < 50 years), present for less than 10 years and in absence of other risk factors	LDL < 100 mg/dL
Low-risk patients (risk of cardiovascular mortality at 10 years < 1%)		LDL < 116 mg/dL

**Table 2 ijms-23-09326-t002:** Indications for initiation or continuation of statin therapy in different categories of high cardiovascular risk patients.

CATEGORY	INDICATION
ACS	Early initiation or continuation of high-dose statin therapy is recommended
PAD	Start or continue statin therapy according to the ESC guideline
HF	Statins should be continued in HFrEF patients already receiving statins for coronary artery disease or hyperlipidemia. Initiation of statins is not recommended for most patients with chronic heart failure
CARDIAC VALVULOPATHIES	It is not recommended to initiate statins to slow progression of aortic stenosis in patients with non-CAD aortic stenosis without other indications for use
STROKE	Statins are recommended for patients with a history of ischemic stroke or TIA
ELDERLY PEOPLE	Starting statin therapy for primary prevention at very high cardiovascular risk may be considered, subject to other factors such as risk modifiers, frailty, estimated benefit over the life course, comorbidities and patient preferences. Statin therapy is generally safe and well-tolerated in these patients, and ongoing treatment should be continued
YOUNG PEOPLE	Starting statin therapy for primary prevention of ASCVD in individuals ≥ 21 years of age with LDL-C ≥ 190 mg/dL is recommended. For individuals 20 to 39 years of age with LDL-C < 190 mg/dL, a lifetime risk assessment is recommended
FAMILIAL DYSLIPIDEMIAS	Children homozygous for FH should be treated as early as possible at the time of diagnosis. Children heterozygous for FH should be initiated at the lowest recommended dose and up-titrated according to the LDL cholesterol-lowering response and tolerability from 8 to 10 years of age
GENDER	If patients at high cardiovascular risk develop cognitive impairment due to statin use and require continued lipid-lowering therapy, the use of less lipophilic statins should be considered
PREGNANCY	Statins should be avoided during pregnancy and discontinued prior to conception due to limited data and information quality
PERIOPERATIVE PERIOD	It is recommended that patients on statins as maintenance therapy continue to use statins in the perioperative period. Statins are indicated for patients with vascular surgery with or without clinical risk factors and should be considered for moderate-risk surgery in patients with at least one clinical risk factor
CHRONIC RENAL FAILURE	Guidelines recommend statins for primary prevention of cardiovascular disease in patients over 50 years of age with eGFR < 60 mL/min/1.73 m^2^. In adults < 50 years with chronic kidney disease, known coronary artery disease, diabetes, previous ischemic stroke, or estimated cardiovascular risk > 10% at 10 years of age, statin therapy should be initiated. Initiation of statin therapy in chronic dialysis patients is not recommended. It should be considered continuing therapy in patients starting dialysis, especially those with ASCVD
MUSCLE DISEASE	Guidelines recommend against starting statin therapy if baseline CK > 4 ULN
HIV	There are no guidelines on statin therapy developed specifically for the primary prevention of cardiovascular disease among HIV-infected patients, but preliminary observational data suggest a potential CVD morbidity and all-cause mortality benefit in routine care
LIVER DISEASE	Statins should be used in these patients if they have a metabolic or cardiovascular indication. In patients with compensated cirrhosis, statins are safe and can be used at conventional dosages. Simvastatin 20 mg and pravastatin 40 mg could be the choice in decompensated patients when there is an indication for statins. Patients with Child–Pugh C have a short-term high liver mortality, which is unlikely to be changed by statins
DIABETES	Statin is the primary target of lipid-lowering therapy in patients with DM2. In light of the well-established cardiovascular risk-reducing effect of statin use, the consensus of experts is that statin therapy should not be discontinued for fear of increasing the risk of diabetes
CANCER	Survival is not affected when statins prescribed for primary or secondary prevention of cardiovascular disease are discontinued in this population
TRANSPLANTED PATIENTS	Statins should be considered as first-line agents in transplant patients, and initiation should be at low doses with careful up-titration and with caution because of potential drug interactions

## Data Availability

Not applicable.

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
