# Peer review of "Statins in High Cardiovascular Risk Patients: Do Comorbidities and Characteristics Matter?"

_ijms, 2022, doi:10.3390/ijms23169326_

Round 1
Reviewer 1 Report
The authors have written an excellent review article on the recommendation of statins in patients with high risk for various major cardiovascular events as well as in special populations with high risk of CVD. They have done a thorough review of the literature, analyzed the information gathered, interpreted them appropriately and made their recommendations.
This reviewer has the following suggestions/recommendations/concerns:
1. They should discuss in more detail the role of statins in treating population with metabolic syndrome (characterized by insulin resistance) but not yet with manifested type 2 diabetes. It should be noted that nearly one-third of US population has metabolic syndrome characteristics. This prevalence may vary across globe, but it is longitudinally increasing in many countries. Therefore, it is important to discuss this population.
2. Extensive formatting of references is required, and accuracy be checked.
3. Please review the last sentence in conclusion, if that is what you really mean to conclude.
4. Please go through the following lines for unclear meaning of sentence/s, incomplete sentence, misplaced word or words, unnecessary capitalization of word, or repeated word.
a) Line 17
b) Lines 52-57
c) Lines 76-78
d) Lines 97-98
e) Lines 183-185
f) Lines 211-217
g) Lines 265-266
h) Lines 314-317
i) Line 348
j) Line 384
k) Line 439
l) Line 461
m) Line 517
n) Line 546
o) Lines 547-549
p) Line 553-554
q) Lines 560-568 (too long a sentence)
r) Line 579
s) Lines 581-583
t) Line 615
Author Response
Reviewer 1
The authors have written an excellent review article on the recommendation of statins in patients with high risk for various major cardiovascular events as well as in special populations with high risk of CVD. They have done a thorough review of the literature, analyzed the information gathered, interpreted them appropriately and made their recommendations.
Authors’ response
We thank the Reviewer for her/his comments. We have revised the manuscript based on the suggestions. We have made every effort to address all queries. Please also find our point-by-point response to the comments.
This reviewer has the following suggestions/recommendations/concerns:
- They should discuss in more detail the role of statins in treating population with metabolic syndrome (characterized by insulin resistance) but not yet with manifested type 2 diabetes.It should be noted that nearly one-third of US population has metabolic syndrome characteristics. This prevalence may vary across globe, but it is longitudinally increasing in many countries. Therefore, it is important to discuss this population.
Authors’ response
We thank the Reviewer for her/his comment. Patients with greater severity of the metabolic syndrome are at higher risk for developing incident T2D due statin use. We added a detailed paragraph on the use of statins in the insulin resistant population, highlighting the increased likelihood of developing diabetes. We also reported the possible mechanisms that can explain the etiology. Please see the 3.12. Statins and diabetes section of the revised manuscript.
- Extensive formatting of references is required, and accuracy be checked.
Authors’ response
We thank the Reviewer for her/his suggestion. We proceeded to format the references as required.
- Please review the last sentence in conclusion, if that is what you really mean to conclude.
Authors’ response
We thank the Reviewer for her/his remark. The purpose of this review was to analyze the indications for initiation or continuation of statin therapy in different categories of patient with high cardiovascular risk, considering their complexity and comorbidities in order to personalize treatment. Physicians should evaluate the patient as a whole and with his comorbidities, considering the risks and benefits of the treatment. Please see the 4. Conclusion section of the revised manuscript.
- Please go through the following lines for unclear meaning of sentence/s, incomplete sentence, misplaced word or words, unnecessary capitalization of word, or repeated word.
a) Line 17
b) Lines 52-57
c) Lines 76-78
d) Lines 97-98
e) Lines 183-185
f) Lines 211-217
g) Lines 265-266
h) Lines 314-317
i) Line 348
j) Line 384
k) Line 439
l) Line 461
m) Line 517
n) Line 546
o) Lines 547-549
p) Line 553-554
q) Lines 560-568 (too long a sentence)
r) Line 579
s) Lines 581-583
t) Line 615
Authors’ response
We thank the Reviewer for her/his remark. We identified and corrected the observations for each line indicated.
Reviewer 2 Report
The authors reviewed the indications for initiation or continuation of statin therapy in different categories of patient with high cardiovascular risk. This review is significant for a clinical practice. For better understanding to the readers, the authors should make summarized Table of the indications and different categories. Moreover, each subsection of clinical studies and meta-analysis needs a separate detailed table including the number of studies, groups, patients, and results with significance with proper references.
Author Response
Reviewer 2
The authors reviewed the indications for initiation or continuation of statin therapy in different categories of patient with high cardiovascular risk. This review is significant for a clinical practice. For better understanding to the readers, the authors should make summarized Table of the indications and different categories. Moreover, each subsection of clinical studies and meta-analysis needs a separate detailed table including the number of studies, groups, patients, and results with significance with proper references.
Authors’ response
We thank the Reviewer for her/his suggestion. A table of the indications for initiation or continuation statin therapy in different categories of patient with high cardiovascular risk was included into the revised version of the manuscript. We also added two different tables (one for patients with cardiovascular diseases and another for special subpopulations) with the most important studies that led us to indicate whether to start or continue the statin therapy. Please see the 4. Conclusion section of the revised manuscript and the Supplemental Tables.
Round 2
Reviewer 2 Report
The authors revised appropriately. No further correction is necessary.